# Approaches for Mitigating Microbial Biofilm-Related Drug Resistance: A Focus on Micro- and Nanotechnologies

**DOI:** 10.3390/molecules26071870

**Published:** 2021-03-26

**Authors:** Harinash Rao, Sulin Choo, Sri Raja Rajeswari Mahalingam, Diajeng Sekar Adisuri, Priya Madhavan, Abdah Md. Akim, Pei Pei Chong

**Affiliations:** 1School of Medicine, Taylor’s University, Subang Jaya, Selangor 47500, Malaysia; harinashraoprecasaroe@sd.taylors.edu.my (H.R.); sekar.adisuri@gmail.com (D.S.A.); priya.madhavan@taylors.edu.my (P.M.); 2School of Biosciences, Taylor’s University, Subang Jaya, Selangor 47500, Malaysia; celinechoo14@hotmail.com; 3School of Pharmacy, Taylor’s University, Subang Jaya, Selangor 47500, Malaysia; sriraj.nachchiyaar@gmail.com; 4Department of Biomedical Sciences, Faculty of Medicine and Health Sciences, Universiti Putra Malaysia, Selangor 43400, Malaysia

**Keywords:** microtechnology, nanotechnology, biofilm-related resistance

## Abstract

Biofilms play an essential role in chronic and healthcare-associated infections and are more resistant to antimicrobials compared to their planktonic counterparts due to their (1) physiological state, (2) cell density, (3) quorum sensing abilities, (4) presence of extracellular matrix, (5) upregulation of drug efflux pumps, (6) point mutation and overexpression of resistance genes, and (7) presence of persister cells. The genes involved and their implications in antimicrobial resistance are well defined for bacterial biofilms but are understudied in fungal biofilms. Potential therapeutics for biofilm mitigation that have been reported include (1) antimicrobial photodynamic therapy, (2) antimicrobial lock therapy, (3) antimicrobial peptides, (4) electrical methods, and (5) antimicrobial coatings. These approaches exhibit promising characteristics for addressing the impending crisis of antimicrobial resistance (AMR). Recently, advances in the micro- and nanotechnology field have propelled the development of novel biomaterials and approaches to combat biofilms either independently, in combination or as antimicrobial delivery systems. In this review, we will summarize the general principles of clinically important microbial biofilm formation with a focus on fungal biofilms. We will delve into the details of some novel micro- and nanotechnology approaches that have been developed to combat biofilms and the possibility of utilizing them in a clinical setting.

## 1. Introduction

The diversity and incidence of infectious diseases have seen an exponential increase over the past 30 years despite advances in the medical field, which have significantly reduced mortality rates [1]. Many researchers have focused their studies on uncovering the complex dynamics underlying the persistence and spread of infectious diseases [2] to obtain a clearer insight of this phenomena. The onset of most infectious diseases caused by bacterial and fungi begins with planktonic cells that adhere to surfaces, eventually forming complex biofilms that protect them from potentially harmful or stressful environments [3].

Biofilms are intricate structures that are often associated with the emergence of various challenges such as horizontal gene transfer, antimicrobial resistance (AMR) and persister cells that result in chronic or recurring infections. They consist of a heterogeneous community of microorganisms that adhere irreversibly to abiotic or biotic surfaces through the production of extra-polymeric material [4]. Bacterial and fungal cells embedded within a biofilm have been observed to be 10 to 1000 folds more resistant to treatment with antimicrobials [5] than their free-living counterparts. These distinctive structures have been identified as the predisposing factor for nosocomial infections, which negatively impacts ongoing treatments and often results in life-threatening situations [4]. The intrinsically complex antimicrobial resistance displayed by biofilms and the limited plethora of new antimicrobial drugs [6] have strongly indicated the need for alternative therapeutics that could combat infections associated with biofilms.

Information gathered on biofilms demonstrated that their composition and mechanisms play an essential role in determining the resistance and virulence exhibited by the constituent microorganisms [7]. Many researchers have directed their focus towards utilizing micro- and nanotechnology as a single or complementary therapeutic to combat biofilms. Micro- and nanotechnology exhibit unique characteristics such as drug carriers as well as the ability to penetrate the extracellular polymer substance (EPS) layers of biofilms that would make them promising antibiofilm candidates. In this review, we will discuss the general principles underlying clinically important microbial biofilm formation with special emphasis on fungal biofilms, the genes involved in biofilm formation and their implications as well as the association of antimicrobial resistance with biofilms. Additionally, we will also elaborate the fundamentals of micro- and nanotechnology, their mechanism of action, fields of utilization, and possible applications as targeted therapy against biofilms.

### 1.1. Biofilms and Their Role in Resistance

Biofilm development is an intricate process that involves multiple sequential steps. Bacteria and yeasts share similar biofilm formation processes, whereas filamentous fungi require several additional steps including the formation of mature filaments and terminal biofilm morphology to establish their biofilms [8]. Generally, biofilm formation involves the attachment of cells, followed by formation of microcolonies, maturation of the biofilms and eventually, dispersion. The initial attachment of cells to surfaces is reversible as these cells are still susceptible to treatment by antibiotics. Subsequent steps are irreversible once the cells have adhered to the surfaces and begin to proliferate. Figure 1a,b depict the biofilm formation of filamentous fungi and yeasts respectively, while Figure 1c depicts bacterial biofilm formation.

Biofilms are intrinsic, dynamic structures that contribute to the development of resistance in microorganisms and play a role in the emergence of AMR [9]. Microorganisms develop AMR as an evolutionary response to survive hostile environments. Factors that have contributed to this phenomenon include human negligence through the uncontrolled prescription of antimicrobials, the inability of antimicrobials to enter the cells such as aminoglycosides and β-lactams or their poor retention abilities as observed with fluoroquinolones and macrolides [10]. Biofilm-related resistance are dependent on several other factors including physiological state, cell density, quorum sensing (QS), extracellular matrix (ECMs), overexpression of drug targets, up-regulation of drug efflux pumps, persister cells, and tolerance.

### 1.2. Biofilms in Healthcare-Associated Infections (HAIs)

According to the Centre for Disease Control [11], one in every thirty-one patients contract at least one type of healthcare-associated infections (HAIs) on any given day. Approximately 60 to 70% of HAIs are attributed to biofilm formation on indwelling medical devices such as catheters, endotracheal tubes, and prostheses [12,13]. Indwelling devices are the ideal surfaces for biofilm formation as fewer number of microorganisms are required for colonization compared to human tissues [14,15,16,17]. *Enterococcus faecium, Staphylococcus aureus*, *Klebsiella pneumoniae*, *Acinetobacter baumannii*, *Pseudomonas aeruginosa*, and *Enterobacter* species (ESKAPE) are some of the most common multidrug resistant bacteria associated with HAIs [18]. Meanwhile, *Candida* spp. are the most common fungal species observed in HAIs [19] with increasing cases caused by non-*Candida albicans* species [20]. The most frequent HAIs reported are central line-associated bloodstream infections (CLABSI), catheter-associated urinary tract infection (CAUTI), ventilator-associated pneumonia (VAP), and surgical site infections (SSI) [21].

### 1.3. Implications of Biofilm-Related HAIs and Possible Treatments

Biofilm-associated infections are not only difficult to diagnose and treat [22] but could also increase mortality rates whilst reducing the efficacy of medical devices. The present approaches to address biofilm-related HAIs emphasizes the importance of preventing infections such as CLABSI and CAUTI through the practice of aseptic techniques and limiting the usage unless necessary or intermittently [23]. Recent studies are exploring the possibility of integrating antimicrobials and indwelling medical devices through coating or impregnation alongside routine aseptic wound dressing procedures or antimicrobial prophylaxis to prevent the formation of biofilms on these abiotic surfaces as observed in the cases of CLABSIs, CAUTIs [23], VAPs [24], and SSIs [25]. These novel techniques to prevent or treat biofilms are promising as they could reduce the incidences of HAIs and the rate of infections associated with indwelling medical devices, ultimately reducing the mortality rate. A variety of nonconventional and new approaches to mitigate biofilms are discussed in a subsequent section of this review.

## 2. Characteristics of Biofilms

The characteristics and composition of biofilms have an indirect impact on their susceptibility towards antimicrobials. We examine the roles of some of these characteristics including the physiological state of microbes within biofilms as well as the ECM components and their contribution towards AMR below.

### 2.1. Physiological State

The physiological state of microorganisms within biofilms can be influenced by environmental factors, which in turn could affect their essential needs and contribute to AMR. Baillie and Douglas [26] reported that *C. albicans* biofilms were highly resistant to Amphotericin B (AMB) when cultured in the absence of glucose and iron. Meanwhile, *C. albicans* biofilms exhibited resistance towards AMB and azole-group drugs when grown under anaerobic conditions [27]. Walters III and co-authors [28] observed a similar finding with *P. aeruginosa* biofilms, which exhibited antibiotics tolerance in low oxygen environment. Other factors including pH, temperature, and environmental pressures can result in morphological changes in the overall biofilm architecture, which might lead to AMR [29].

### 2.2. Extracellular Matrix (ECM)

Extracellular matrix (ECM) is one of the main characteristics of biofilms that provides protection from hostile environments such as antimicrobial agents and host immune system [30]. It serves as a platform for QS, provides mechanical or structural stability, prevents the penetration of antimicrobial agents, as well as aids in the movement of nutrients and energy in and out of the biofilm [31]. A wide range of conventional antimicrobials such as azoles currently utilized are not able to exert their effects appropriately due to their inability to penetrate the extracellular polymeric substances (EPS) layer effectively.

ECMs are generally composed of polysaccharides, proteins, nucleic acids, phospholipids, lipids, amyloid fibers, humid substances, and extracellular DNA (e-DNA) (30. In fungi, β-1,3 glucan, β-1,6 glucan and mannan are carbohydrate components of the ECM and major constituents of fungal cell walls [30]. Meanwhile, in bacteria, cellulose [32] and other species-specific carbohydrates such as the staphylococcal polysaccharide intercellular adhesin isolated from *S. aureus* [33] are described as the carbohydrate components of their ECMs [34]. ECM components may vary among microorganism species, which results in variations in resistance. However, its role in AMR is undeniable. Table 1 describes the biofilms matrix components of several fungi commonly isolated from (HAIs).

Recent studies have shown that extracellular DNA (eDNA) found in both bacterial and fungal biofilms [41] is an important component of ECMs as it assists in biofilm structural integrity and maintenance. Evidence also suggests that degradation of eDNA causes the collapse of biofilm architecture while the addition of exogeneous DNA promotes biofilm growth [42].

## 3. Factors Contributing towards Drug Resistance

Biofilm-related resistance are dependent on several factors including cell density, QS, ECMs, overexpression of drug targets, up-regulation of drug efflux pumps, persister cells, and antimicrobial tolerance.

### 3.1. Cell Density

Fungal biofilms are densely populated with hyphae, pseudohyphae, and yeasts, that are arranged in an orderly system for important functions such as nutrient perfusion, waste expulsion and water channels [29]. Meanwhile, bacteria biofilms consist of either one or more microorganisms [43] arranged in an organized yet complex fashion to facilitate essential functions. Cell density is described as an important factor in resistance as it was previously observed that the planktonic cells and biofilms of *C. albicans* had reduced susceptibilities towards the azole-class drugs, as well as AMB and caspofungin when cell densities were increased from 1 × 10^3^ to 1 × 10^8^ cells/mL [44]. Moreover, cell density has been attributed as one of the main factors underlying AMR as planktonic cells and biofilms of *E. coli* and *S. aureus* with similar cell density demonstrated the same level of susceptibility towards several antibiotics [45].

### 3.2. Quorum Sensing (QS)

Quorum sensing (QS) is defined as the ability of microorganisms to communicate and coordinate their activities through the secretion of signaling molecules known as quorum-sensing molecules (QSMs), in a population-dependent manner [46]. Turan and colleagues [47] have described QS as a key player in several cellular activities such as pathogenic gene expression, toxin production and extracellular polysaccharide synthesis. It also imparts an important regulatory role in the process of drug efflux pumps and the formation of microbial biofilms [47].

There are approximately four known QSMs identified in fungi, which are farnesol, tyrosol, phenylethanol, and tryptophol [48]. Meanwhile, QSMs in bacteria are more distinctive depending on whether they are Gram negative or Gram positive. The autoinducer, N-acyl homoserine lactones (AHL) as well as other molecules synthesized by S-adenosylmethionine are known QSMs of Gram negative bacteria [49] whereas, autoinducing peptides are well-documented Gram positive QSMs that are responsible for interacting with a two-components histidine kinase signal transduction system [50,51]. Both Gram negative and Gram positive bacteria are able to secrete and detect the QSM, autoinducer 2 [52]. Association of QSMs with resistance was demonstrated by Cao et al. [53] who reported that two antifungal resistance genes, Ca*FCR1* and Ca*PDR16* in *C. albicans* were upregulated in the presence of exogenous farnesol. Likewise, a QSM, AHL N-3-hydroxy-dodecanoyl-homoserine lactone (N-3-OH-C12-HSL) notably restored the function of antibiotic resistant genes in *A. baumanii* [54].

### 3.3. Upregulation of Drug Efflux Pumps

Efflux pumps are transport proteins responsible for the removal of antimicrobial agents particularly, azole-groups drugs [29] from within the cells. Many studies have shown that the expression of efflux pumps is upregulated in bothbacterial and fungal biofilms upon exposure to antimicrobial agents, which contributes to AMR during infection. Efflux pumps in fungal species belong to two main classes of transport proteins, which are ABC (ATP-binding cassette), a primary active transporter that uses energy from hydrolysis of ATP to drive the efflux of antifungal agents and MFS (Major Facilitator Superfamily), a secondary active transporter that uses the electrochemical gradient of protons across the plasma membrane of efflux substrates [55]. Table 2 shows the genes encoding the efflux pumps expressed in fungal biofilms.

The importance of the drug efflux pumps was observed in a study involving a set of isogenic *C. albicans* strains that were lacking in one or more *Cdr1p*, *Cdr2p*, and *Mdr1p* pumps whereby, the strains lacking the pumps were more susceptible to fluconazole at the early stages of biofilms compared to the wildtype [60]. A similar finding was noted with *C. tropicalis* as the upregulation of *MDR1* led to increased resistance towards AMB. However, the researchers eventually ruled out the role of this gene in resistance and surmised that the increased expression of *MDR1* may be a protective mechanism. In fact, a finding by Arana et al. [61] indicates that *C. albicans* exhibited a resistance towards oxidative response instead of the antifungal itself as noted when exposed to subinhibitory concentrations of fluconazole. On the other hand, the expression of the efflux pump, *AfuMDR4* were notably upregulated in vivo upon exposure to voriconazole [62].

Bacterial efflux pumps can be categorized into six different groups, which are the ABC superfamily [63] and MFS [64] as seen in fungi species, multidrug and toxic compound extrusion, or MATE [65], small multidrug resistance or SMR [66], resistance-modulation-division (RND) superfamily [67] and drug metabolite transporter (DMT) superfamily. RND are found in most clinically relevant Gram negative bacteria [67] whereas, ABC and MFS are the most common pumps found in Gram positive bacteria, with the latter found ubiquitously in various microorganisms [64].

Unlike fungi, bacterial efflux pumps have been extensively studied and their role in AMR is undeniable. A study on *Stenotrophomonas maltophilia* described an ABC type efflux pump, MacABCsm as the main contributor of the resistance towards antibiotics from the macrolides, aminoglycosides and polymyxins groups [68]. A prevalence study conducted over three years to determine fluoroquinolone resistance in clinically isolated *E. coli* observed a *gyrA* mutation in more than 50% of the species that have been associated with organic solvent tolerance. This phenomenon may lead to the overexpression of efflux pumps or further mutation of this gene overtime, which would indirectly contribute to AMR [69]. Additionally, mutant strains of *P. aeruginosa* that overexpressed or could not express efflux pumps exhibit increased susceptibility to antibiotics when compared to the wildtype [70].

### 3.4. Point Mutation and Overexpression of CDR1, ERG11

Point mutations in the genes encoding targeted enzymes such as *ERG11* as well as in the efflux pump-encoding gene, *CDR1* in fungi increase tolerance towards antimicrobials, which could contribute to AMR. Several reports have detailed the point mutations and increased expression of *CDR1* genes in *C. albicans* [71] and *C. glabrata* [72] that were associated with azole resistance in clinical isolates. The *ERG11* gene is one of the most studied genes in fungi that encodes for the enzyme, lanosterol 14 α-demethylase, which is a target enzyme for azole-group drugs. Point mutations of *ERG11* gene alters the azole-binding site [73] and also leads to the overexpression or upregulation of the *ERG11* genes, which leads to reduced affinity as well as unspecific binding [74]. Common mutations in the *ERG11* gene in *C. albicans* which are associated with azole resistance have been identified as *S405F*, *Y132H, R467K*, and *G464S* [29]. On the other hand, mutations in the *CYP51A* gene which leads to changes to ergosterol pathway that confer azole resistance were also observed in *A. fumigatus* [74]. Spettel and co-workers [75] had analyzed the mutations in azole-resistant, echinocandin-resistant as well as multi-resistant strains of *C. albicans* and *C. glabrata* using next-generation sequencing, and found over fifty different missense mutations in the various genes including *ERG11*, *ERG3, TAC1*, and *GSC1* (*FKS1*) in *C. albicans* and *ERG11, CgPDR1, FKS1*, and *FKS2* in *C. glabrata*.

*Candida* species triggers a feedback mechanism that upregulates the overexpression of *ERG11* in response to ergosterol depletion when exposed to azole drugs [74], which would increase the concentration of antifungal required and contribute to resistance. Other genes such as *ERG1*, *ERG3*, *ERG7*, *ERG9*, and *ERG25* are also reported to play important roles in the resistance towards antifungals [76]. Rodrigues and colleagues [77] noted that *ERG3*, *ERG6*, and *ERG11* were upregulated in *C. glabrata* biofilms in the presence of fluconazole and voriconazole. Similarly, Cd*ERG3* and Cd*ERG25* genes of *C. dubliniensis* biofilms were upregulated following incubation with fluconazole [76]. Rossignol and co-authors [78] discovered that *ERG11* gene were upregulated in *C. parapsilosis* biofilms compared to those in planktonic cells. Meanwhile, the exposure of *C. albicans* biofilm to fluconazole exhibited upregulation of *ERG1*, *ERG3*, *ERG11*, and *ERG25* [79]. A majority of the studies on *ERG11* involve planktonic cells, and hence more studies would have to be conducted on biofilms to determine if these results obtained in the former are replicated in the biofilms.

Likewise, bacterial species have been reported to have their fair share of mutations that contribute to their resistance. Six of the drug resistant genes which are *gidB*, *gyrA*, *gyrB*, *rpoB*, *rpsL*, and *rrs* have been observed to undergo similar mutation patterns in bacteria [80].

### 3.5. Presence of Persister Cells

Persister cells are dormant cells within biofilms that are able to tolerate high concentrations of antifungal agents [29] and cause recurrent infections. Persister cells are found in both planktonic cells and biofilms in bacteria, whereas they have only been observed in biofilms for fungal species [30]. They are phenotypic variants of their wild-type counterparts with identical genetic profile as antimicrobial susceptible cells [30,81] and thus, lack heritable resistance mechanisms. Persister cells formation is independent of biofilm formation as it was previously detected in a consistent level even in mutant strains such as *efg1*Δ/Δ, *cph1*Δ/Δ, and *mkc1*Δ/Δ, that were defective in biofilm formation [82]. A recent study on *E. coli* has indicated that persister cells may exhibit a long retention effect after it has been secreted from biofilms [83].

The presence of persister cells is only observed when microbicidal agents are used and are reported to reduce their susceptibility to antimicrobial agents as well as increase their survivability. LaFleur and co-workers [82] detected persister cells in *C. albicans* biofilms treated with fungicidal agent, AMB, which later gave rise to biofilms containing subpopulations of persister cells. Similarly, Al-Dhaheri and Douglas [84] identified the presence of persister cells in the biofilms of *C. albicans*, *C. krusei*, *C. parapsilosis* treated with different concentration of AMB using fluorescein diacetate staining. Persister cells (1 to 2%) were observed in *C. albicans* biofilms when treated with 0.6 to 2.4 µM of miconazole [85]. As *Candida* species have developed resistance towards a wide range of azoles including fluconazoles and miconazole, this family of antifungals may not be suitable to deduce the presence of persister cells [86]. On the contrary, AMB resistance is uncommon, which indicates that cells that survive treatment with this antifungal are likely persister cells and not resistant [86,87].

Persister cells have also been described in bacteria and their presence has been associated with AMR. The exact mechanism behind these phenotypes is uncertain however, as with fungi, it has been hypothesized as a stress response in the presence of antibiotics or as an adaptive evolution in the form of spontaneous persistence in order to survive [88].

### 3.6. Antimicrobial Tolerance

Antimicrobial tolerance is the ability of microorganisms to survive transient exposure to high concentrations of antibiotics that exceed the minimum inhibitory concentration (MIC) [86], which could lead to treatment failure. Delarze and Sanglard [89] have described tolerance as an outcome of epigenetics compared to resistance, which depends on alterations in the microorganisms’ DNA sequences. On the contrary, Brauner and colleagues [90] surmised that genetic mutations and environmental factors may contribute to tolerance.

Many studies have shown that poor growth conditions also lead to tolerance [90]. Three meropenem tolerance genes, which are BTH_I0069, *ldcA*, and *deg*S have been identified in *Burkholderia* species including *Burkholderia pseudomallei and Burkholderia thailandesis* when grown under three conditions: in stationary phase, in the absence of carbon and in biofilms. Tolerance mechanisms are also specific to the antimicrobial agents used and experimental conditions used [91]. Similarly, Lederberg and Zinder [92] have isolated *E. coli* mutants that survived penicillin exposure in the absence of an amino acid.

Antifungal tolerance is implicated by two pathways, which are the Ca-calmodulin-calcineurin and cAMP-protein kinase A pathways. These signal transduction pathways play pivotal roles in fungal adaptation to hostile environments such as nutrient imbalance, pH, oxygen concentrations, temperature, and host immune response [93]. Calcineurin is a Ca^2+^-calmodulin-activated serine/threonine-specific protein phosphatase that is important in cellular response to stress [94]. Many studies have reported that calcineurin contributes to azole resistance in fungal biofilms especially in *Candida* and *Aspergillus* species [29,95]. Uppuluri and co-workers [96] have documented that combination of calcineurin inhibitor and fluconazole treatment provides synergistic effects in eradicating *C. albicans* biofilms via in vitro and in vivo rat catheter model experiments. Meanwhile, the disruption of Ca^2+^-mediated calcineurin signaling pathway decreased the formation of biofilm by *Aspergillus niger* [95].

cAMP-Protein kinase A (PKA) pathway is important in cellular function in response to glucose in pathogenic yeasts such as *Saccharomyces cerevisiae*, *C. albicans*, *Cryptococcus neoformans*, and *Aspergillus fumigatus* [97]. cAMP-PKA pathway is also an antagonist of calcineurin stress response pathway. In *C. albicans* biofilms, the cAMP-PKA pathway help cells recover and commence growth after stressful conditions such as fluconazole exposure. *CDC35* is the cAMP-associated protein that is responsible for fluconazole tolerance in *C. albicans* and inhibiting these genes results in hypersusceptibility to azoles and terbinafine [94]. cAMP-PKA pathway also regulates *S. cerevisiae* biofilm formation and maintenance [98] as noted through the detection of the genes *Phd1*, *Ash1*, *Mga1*, and *Sok2*. Echinocandins resistance in *A. fumigatus* was reduced by inhibiting Hsp90, a molecular chaperone that regulates temperature-dependent fungal morphogenesis through repression of cAMP-PKA signaling [99].

## 4. Potential Alternative Therapeutics for Biofilm Mitigation

Although antibiotics and antifungal drugs are the mainstay for treating microbial infections, alternative therapeutic approaches have been studied for their effectiveness to combat biofilm-related drug resistance. The following sections describe the various alternative therapeutics under development or currently in use for mitigating biofilm resistance.

### 4.1. Antimicrobial Photodynamic Therapy (aPDT)

Antimicrobial photodynamic therapy (aPDT) interchangeably referred to as photodynamic activation (PDI) or photodynamic antimicrobial chemotherapy is an alternative therapeutic mode for localized biofilm infections. The principle behind aPDT is that photosensitizer (PS) dyes generate sufficient reactive oxygen species (ROS) from molecular oxygen upon exposure to a light source at a specific wavelength to trigger oxidative stress followed by microbial cell death without exerting toxic effects in the host [100]. aPDT is a result of synergism between three components which are non-toxic PS, molecular oxygen, and visible light [101,102].

Numerous in vitro and in vivo studies have shown that aPDT is effective in eradicating biofilms formed by bacteria and yeasts [103]. Junqueira and co-workers [104] reported that aPDT involving ZnPc (cationic nanoemulsion of zinc 2-,9-,16-,23-tetrakis(phenylthio)-29H, 31H-phthalocyanine) led to a reduction in CFU ranging from 0.33 to 0.85 log_10_ for *Candida* species, 0.84 log_10_ for *K. ohmeri* and 0.85 log_10_ for *T. mucoides*. Meanwhile, methylene blue-mediated aPDT reduced the CFUs of biofilms formed by *Trichophyton rubrum*, *Trichophyton mentagrophytes*, *Microsporum gypseum* [103], mixed biofilm consisting of *P. aeruginosa* and methicillin-resistant *Staphylococcus aureus* (MRSA) [105] as well as *E. faecalis* biofilms [106].

### 4.2. Antimicrobial Lock Therapy (ALT)

Antimicrobial lock therapy (ALT) is an alternative therapy used to treat biofilm-related infections associated with medical devices such as catheter. The theory behind ALT is that the instillation of antimicrobial agents, which exceeds the planktonic MIC by 100- to 1000-fold within an intravascular catheter lumen will allow it to stay ‘locked’ over a certain amount of time and lead to a continuous release of the antimicrobial agents [107,108].

*Candida* species is the third most common cause of CLABSIs and the leading species associated with biofilm-related infections [107]. Schinabeck and co-workers [109] have demonstrated the effectiveness of ALT containing liposomal AMB in eradicating *C. albicans* biofilms from catheters that were surgically placed in New Zealand white rabbits after seven days of treatment. ALT containing 0.25 mg/mL of caspofungin prepared in pyrogen-free sterile saline was observed to treat central venous catheter (CVC)-associated candidiasis caused by *C. albicans* in mice [110]. Meanwhile, the administration of ALT in combination with trimethoprim, EDTA, and ethanol (B-lock) reduced *C. albicans*, *C. krusei*, *C. glabarata*, *C. parasilosis*, and *C. tropicalis* by 99.9%; and *E. faecalis*, MRSA as well as methicillin-susceptible *S. aureus* isolates by 50% [111]. The findings from several clinical studies indicate that ALT is an effective method for eradicating microbial infections without the need to remove the catheters. It was also concurred that ALT exhibited better therapeutic effects when used in combination with pre-existing antimicrobials.

### 4.3. Antimicrobial Peptides (AMPs)

Antimicrobial peptides (AMPs) are short (<100 amino acids), positively charged and amphiphilic peptides that are capable of interacting with biological membranes [112]. They form pores to disrupt the cell membrane and inhibit the synthesis of cell wall, enzymes, nuclei acids and proteins by penetrating the cytoplasm of microorganisms [113]. AMPs exhibit a wide range of properties including antibiofilm, anti-cancer, antimicrobial and immunomodulatory properties [114].

AMPs have broad spectrum of inhibitory activities against bacteria, fungi, parasites and viruses [113,114]. A synthetic short cationic AMP, Peptide 1037 which was composed of 9 amino acids exhibited antibiofilm properties against *P. aeruginosa* and downregulated biofilm associated genes [115]. In another study, de Alteriis and colleagues [116] demonstrated that membrane penetrating peptide gH625 and its analogue, gH625-GCGKKKK effectively eradicated infections caused by polymicrobial biofilms, *C. tropicalis*/*Serratia marcenscens* and *C. tropicalis*/*S. aureus* at concentrations lower than the MIC of their planktonic cells. A synthetic kaxin peptide, dF21-10K, successfully eradicated *C. albicans* and *C. tropicalis* biofilms at concentrations 10-fold higher than the MIC [117]. Furthermore, synthetic defensin-like peptides such as α-defensin-3, β-defensin-1, β-defensin-3, and PG-1 exerted inhibitory and antibiofilm activities towards *C. neoformans* biofilms [118].

### 4.4. Electrical Method

Electrical method is a potential antibiofilm therapeutic that involves the application of direct current (DC) to reduce or prevent biofilm formation on indwelling medical devices. Biofilms are initiated when the attraction forces between microorganisms and surface area are greater than the repulsion forces [17]. The application of DC will exacerbate the repulsion electrostatic forces to disrupt the adherence of microorganisms as well as alter the physical conditions such as pH and temperature to further impede biofilm formation [119]. This method is promising as it will not induce resistance and has a low toxicity. Previous findings demonstrated that DC could eradicate the biofilms of *S. aureus* [120] and *S. epidermidis* [121]. Ruiz-Ruigomez and co-workers [119] noted a reduction in *E. coli*, *P. aeruginosa*, *S. aureus*, and *S. epidermidis* by 1, 2, 1, 2 log_10_ CFU/cm^2^, respectively, after inducing 500 µA for 12 h while *C. albicans* showed a 2 log_10_ CFU/cm^2^ after 24 h.

### 4.5. Antimicrobial Coatings

Aside from electrical method, antimicrobial coatings are promising alternative therapeutics to eradicate biofilm-related infections. Medical devices associated with biofilm formations are coated with antibiofilm layers to prevent the adherence of microorganisms to surfaces [17]. These coated surfaces serve as contact killing surfaces [122] while their surface charges will discourage the adherence of microorganisms as observed with antifouling materials. De Prijck et al. [123] observed that immobilized quaternized polyDMAEMA (dimethylaminoethylmethacrylate) and polyethyleneimine on polydimethylsiloxane (PDMS) materials inhibited the formation of *C. albicans* biofilms. Meanwhile, it has been reported that surfaces coated with the naturally-occurring polysaccharide chitosan (partially deacetylated poly *N*-acetyl glucosamine) exhibit antibiofilm activities towards *C. albicans, K. pneumoniae*, *P. aeruginosa, S. aureus*, and *S. epidermidis* [124].

## 5. Nanotechnology and Microtechnology in Antimicrobial Resistance

The emergence of new resistance mechanisms is steadily rising with increasing reports of multidrug resistance across a wide range of microorganisms. In 2019, CDC reported that more than 2.8 million individuals contract antibiotic-resistant related infections yearly in the US alone, with 35,000 of these cases resulting in death [125]. The limited availability of new antimicrobial therapeutics due to the complex process, high cost, time required for clinical trials and approval have resulted in inadequate alternative therapeutics to treat infections caused by these microorganisms [126]. Many researchers have resorted to exploring alternative methods to overcome the challenges presented with biofilms and their role in AMR whilst improving the efficiency of conventional antimicrobials [126]. The following sections elaborate the micro- and nanotechnology-based platforms for mitigating biofilm-associated resistance.

### 5.1. Nanotechnology and Its Mechanisms to Mitigate Biofilms

Nanotechnology have been described as an appropriate alternative or complementary therapeutic to combat biofilms as they can penetrate the outer EPS membrane to deliver antimicrobials directly to the targeted cells or pathogens without fear of degradation [10]. They can improve the release and retention time of antimicrobials intracellularly for a desired amount of time at a therapeutic concentration [127]. Moreover, they have demonstrated antimicrobial properties that could be utilized to develop novel structures such as coatings or antimicrobial surfaces [128].

Nanomaterials have unique physiochemical properties including surface charges and solubility that contribute to controlled biodistribution, intracellular uptake and clearance [126]. Their minute size contributes to their large surface area-to-volume ratio, which gives them improved loading efficiency and differentiates them from their bulkier counterparts [129,130,131]. Meanwhile, their surface charges and zeta potential determine the interaction between nanomaterials and biofilms or cellular membranes, which subsequently affects internalization or intracellular uptake. Interestingly, nanomaterials are reported to enhance the effects of some existing antimicrobials whilst reducing the risk of stimulating new resistant mechanisms by utilizing different mechanisms of action that render current resistance strategies such as efflux pump and thickening of the cell wall redundant [132]. They also exhibit promising antibiofilm properties such as the generation of ROS and inducing cell death by direct contact with the cells.

### 5.2. Types of Nanomaterials

Nanomaterials are categorized as inorganic or organic based on their composition and properties. Inorganic nanomaterials such as gold or magnetic nanoparticles are biocompatible, hydrophilic, and non-toxic with higher stability compared to organic materials [133]. Meanwhile, organic nanomaterials such as liposomes and polymeric nanoparticles exhibit high biocompatibility and biodegradability that make them ideal candidates for clinical applications [134]. Both groups of nanomaterials have their advantages and disadvantages, which could be beneficial if they are applied appropriately. Figure 2 illustrates the main types of novel nanomaterials which exhibit antibiofilm properties.

The present review describes some upcoming novel nanomaterials that have exhibited the potential to combat biofilms and their possible mode of action. Some of these nanomaterials have been extensively utilized in the industry such as for the treatment of wastewater or biofouling, hence further studies will have to be conducted to establish their characteristics such as biocompatibility and safety before they can be utilized for clinical purposes.

#### 5.2.1. Quantum Dots (QDs)

Quantum dots are luminescent, colloidal semiconductor crystals about 2 to 20 nm in size that were named after the ‘quantum confinement’ effect [135]. They exhibit unique tunable broad absorption characteristics with narrow emission spectra, which are continuous and size dependent [136]. QDs can simultaneously emit discrete colors with varying wavelengths upon exposure to broad continuous excitation light due to quantum size effects [137]. The brightness emitted by QDs are reported to be 10 to 100 times greater than most organic dyes or proteins [136]. They are extremely stable against photobleaching and have occasionally been utilized as alternatives for traditional fluorescent dyes or markers [138]. They have been proposed as an ideal tool for various biomedical applications such as detection, imaging and targeting [135] due to their photoluminescence and surface functionalization capabilities [139]. Findings on QDs functionalized with various charges and head functional groups indicate that proper engineering of surface properties would aid their penetration and direct their distribution within a biofilm [140], which implies that QDs can be programmed to deliver drugs directly into a biofilm at a specific site.

QDs have often been used for identification and visualization of structures but recent studies have also shown its antibiofilm potential. A glass surface coated with copper oxide QDs exhibited efficient bactericidal and antibiofilm activities towards *E. coli* as well as *S. aureus*, and these activities were associated with their combined ability to generate massive ROS intracellularly [141]. Meanwhile, graphene oxide (GO) QDs covalently functionalized with polyvinylidene fluoride membrane inhibited the growth and prevented biofilm formation of *E. coli* through contact-inhibition [142]. An experiment by Garcia and co-authors [143] to develop an antimicrobial resin containing QDs conjugated with zinc oxide (ZnO) was observed to decrease the biofilm formation of *Streptococcus mutans* NCTC 10449 by 50% with no signs of cytotoxicity. The combined effect of these two nanoparticles is speculated to enhance their individual antimicrobial activities while reducing their cytotoxicity, thus making it a possible strategy to combat biofilms. A similar finding was observed in their follow-up study using QDs with tantalum oxide whereby reduction in *S. mutans* biofilm formation was noted on the experimental adhesive resin [144].

In a recent study, synthesized carbon QDs reduced the number of cells that could adhere to a polyester surface by either inhibiting quorum-sensing processes or the electrostatic attractions between the QDs and the test pathogen, *C. albicans* MTCC 227. This resulted in the disruption of the adhesin-mediated interactions between *C. albicans* and the plate surface [145]. Additionally, photoexcitation of GOQDs conjugated with curcumin triggered antimicrobial effects towards both mixed species planktonic cells (93% reduction in colony counts) and mixed biofilms (76% reduction) of the perio-pathogens, *Aggregatibacter actinomycetemcomitans* ATCC 33384*, Porphyromonas gingivalis* ATCC 33277, and *Prevotella intermedia* ATCC 49046 through the combined effects of ROS and downregulation of genes involved in biofilms production [146]. Graphene QDs have the ability to disperse amyloid-rich biofilms by competitive assembly with amyloid peptides as seen with the biofilm of *S. aureus* [147].

Current findings have expressed QDs as an appropriate candidate for a wide range of biomedical applications due to their unique characteristics. However, QDs have not been widely applied due to the high cost of precursors and poor reproducibility [145]. Most of the studies are currently in the in vitro stages, indicating the need for in vivo studies to determine if the findings are similar.

#### 5.2.2. Carbon-Based Nanoparticles

Carbon-based nanomaterials such as carbon nanotubes (CNTs), fullerenes and graphene as well as their derivatives have attractive properties that make them potential candidates for clinical applications including drug delivery, imaging, and diagnosis. They have exhibited electrical, optical, and mechanical properties that are unique to each member of the carbon family [148].

#### 5.2.3. Carbon Nanotubes (CNTs)

Carbon nanotubes are small, cylindrical, hollow carbon sheets engineered with opened or closed ends [149], which are one atom thick and derived from rolled graphene planes [150]. They have also been described as long, tubular fullerene structures with hexagonal carbon walls tipped with pentagonal rings [149,151]. Graphene, the main component of CNTs, contribute to the sp^2^ hybridization associated with CNTs [148].

CNTs have been garnering more interest over the past decade as they exhibit ideal drug delivery characteristics such as simple hydrophobic interaction, π-π stacking interaction, electrostatic adsorption, covalent bonds as well as high adsorptive capacity (hollow cylinders) that enables them to cross the cell membrane without any changes to the drug [152]. The inner volume of the CNTs is large enough to accommodate both low and high molecular weight drugs as well as hydrophilic and lipophilic drugs. They have been observed to improve the efficiency and efficacy of the drug by shortening the delivery time [153]. Similarly, they can be utilized for controlled and extended drug release [154,155]. Multiwalled carbon nanotubes (MWCNTs) modified with titanium alloy, TiAl6V4 surfaces, and impregnated with rifampicin reduced the biofilm formation of *S. epidermidis* ATCC 35984 over a period of 10 days through the slow release of the antibiotic attributed to CNTs [156].

Previous findings on CNTs combined with another compound demonstrated complementary or improved antimicrobial properties towards test pathogens in a dose-dependent manner. MWCNTs conjugated with nitrogen, fluorine, and phosphorus was observed to inhibit *B. subtilis, E. coli*, *K. pneumoniae*, and *P. aeruginosa* biofilms by 73.23%, 77.93%, 70.59%, and 79.85% at 250 µg/mL, respectively, while MWCNTs conjugated with nitrogen, fluoride and boron at the same concentration was observed to reduce the same bacterial strains by 82.53%, 80.98%, 76.83%, and 77.41% [157].

Polyethylene (PE) and MWCNTs incorporated composite surfaces led to the formation of thinner biofilms. PE–2% MWCNTs were able to reduce the cell count of *Mycobacterium smegmatis* and *Pseudomonas fluorescens* by 21% and 54%, whereas PE–4% MWCNTs reduced *M. smegmatis* and *Pseudomonas fluorescens* by 29% and 89.3%, respectively [158]. As previously reported, the inhibitory effects exerted by PE-MWCNTs composite have been attributed to the reduction of surface charges and the presence of CNTs, which leads to less accumulation of nutrients on the surface.

Based on the principles of aPDT, rose bengal functionalized with CNTs demonstrated a 64.94 ± 2.91% reduction in biofilm production with a decrease in cell viability by 61.19 ± 2.05% and EPS by 50.19 ± 2.03%, which was notably higher upon exposure to light [159]. A similar study involving toluidine blue conjugated with CNTs reduced biofilm formation of *P. aeruginosa* and *S. aureus* by 69.94 ± 2.90 and 74.54 ± 3.77%, respectively with cell viability dropping by 56.64 ± 2.15 and 28.82 ± 1.51% whilst the EPS production decreased by almost half [160].

Another study by Anju and co-authors [161] to determine the antibacterial properties of malachite green conjugated to carboxyl functionalized MWCNTs was reported to reduce the biofilm formation by 60.20 ± 3.86% for *P. aeruginosa* and 67.59 ± 3.53% for *S. aureus* with a 61.53 ± 3.86% and 62.54 ± 3.00% decrease in cell viability, respectively, after treatment.

On another note, nanoporous alumina substrates that were modified with CNTs showed a reduction in biofilm adherence by *E. coli* ATCC 25922 and *S. aureus* ATCC 25923 while enhancing dental pulp stem cell growth. MWCNTs increased the surface roughness and hydrophilicity of the alumina substrate, which reduced the adherence of bacterial cells by limiting the interactions between the modified surface and the bacteria cells. Interestingly, bacterial cells that manage to adhere onto MWCNTs will sustain damage to the membrane walls. In a similar study, MWCNTs modified nanoporous solid state membranes exhibited effective bacterial killing abilities of >98% in vitro that eventually led to their desorption from the surface [162].

Additionally, vertically aligned MWCNTs that were 470 and 540 µm in length, grown on silicone surfaces exhibited the ability to reduce the volume and manipulate the architecture of the biofilms formed. The authors speculated that the lengthier MWCNTs proved to be more effective as they are more flexible and can oscillate, hence creating an unstable platform which impedes the bacterial cells in the stage of initiation of biofilm production [163].

The findings gathered from past studies indicate that CNTs could serve as an alternative therapy to target biofilms. Their antimicrobial properties have been attributed to their direct contact killing and surface charges. Similarly, their combined effect improves the effects of compounds used in combination whilst strengthening the compounds they are integrated with. However, CNTs have demonstrated certain undesirable characteristics such as agglomeration and cell toxicity when not properly functionalized as well as color changes when utilized with dental resin, that would have to be addressed before they can be utilized for clinical usage.

#### 5.2.4. Fullerenes

Fullerenes are molecules composed of carbon atoms that are arranged in a spherical (Buckyball), ellipsoidal or other various structures [164], which exhibit tunable properties due to their π-electron nature. The curvature of their structure contributes to their rich chemical behavior and enables the synthesis of a wide range of derivatives, making them a versatile building block for various fields of study [165]. Fullerenes have been reported to exhibit antiviral, antioxidative activities, photosensitivity and can function as drug or gene carriers. They are biocompatible and have been shown to penetrate the skin, which makes them an ideal alternative antimicrobial therapy [166].

Functionalized fullerenes have demonstrated antibiofilm and antimicrobial activities as well as improved physical properties of substances. Polystyrene and fullerene composites exposed to plasma exhibited optimal antibacterial as well as antibiofilm activities against clinical isolates of *P. aeruginosa* KT337488 and *S. aureus* KT337489 at 30 min [167]. The presence of functional groups O_2_ and N_2_ enhanced the antibacterial properties exhibited by the composites. Findings from this study indicates that fullerenes could be used in addition to polystyrene to prevent bacteria adhesion and the formation of biofilms. A study conducted by Darabpour et al. [168] demonstrated that fullerenes functionalized with sulfur were able to reduce the biofilm formation of a multidrug resistant strain of *P. aeruginosa* by 92.2% at a minimum biofilm inhibitory concentration (MBIC) of 1 mg/mL, in a concentration dependent manner. The conjugates’ minimum biofilm eradication concentration (MBEC) against pre-established, 1 to 2 days and 3 to 4 days old biofilms were 2 mg/mL and 4 mg/mL, respectively, which has been attributed to the thick EPS layer in older biofilms. A closer examination of the biofilm using field emission scanning electron microscopy (FESEM) after exposure to MBEC showed disrupted structures with reduced cell density.

Fullerenes have been observed to exert antibiofilm and antimicrobial activities when used in combination with other compounds. However, the exact underlying mechanisms that contribute to these antimicrobial activities remains unknown. It is speculated to be due to their singular or combined effects on the microorganisms’ respiratory chain, disruption, or interaction with the cellular membrane [169]. Current data suggest that these biocompatible nanoparticles could make the ideal drug delivery vehicle for targeting clinical applications with sufficient in vivo data.

#### 5.2.5. Graphene

Graphene is a single layer of sp^2^ hybridized carbon atoms arranged in a two-dimensional form and covalently bonded in a hexagonal lattice. They are considered the thinnest yet strongest in existence [170,171]. They are the building block for a wide range of carbon materials such as Bucky ball, graphene QD and one dimensional CNTs [172]. They exhibit remarkable electrical, mechanical, and optical properties. The current challenge faced with utilizing graphene for clinical purposes is the contradictory reports observed with their biocompatibility [173]. Follow up studies on their safety have to be conducted before they can be applied clinically.

Bregnocchi et al. [174] demonstrated that graphene nanoplatelets (GNP) were able to lower the CFU count of *S. mutans* ATCC 25175 at 0.2%, which inevitably caused a 56% reduction in biofilm formation. GO nanosheets were observed to exert antibiofilm activities towards *E. coli* and *P. aeruginosa* in a concentration and time dependent manner [175]. A Foley-type catheter coated with graphene and graphene with silver demonstrated the lack of visible biofilms on the surface after incubation [176]. A closer examination of the coatings through microscopy noted a few *S. epidermidis* cells adhered to the surface, with no biofilm formed despite prolonged culturing. It is speculated that some of these cells might be mechanically trapped on the surface due to the higher roughness.

Titanium is a material commonly utilized in implants. However, it is prone to bacterial colonization on the surface. A study conducted on graphene-coated titanium was observed through confocal microscopy to reduce bacterial biofilms by disrupting the biofilm structure [177]. Interestingly, the coating did not induce cell death, instead it appeared to prevent adhesion, which the authors have inferred to be through the action of surface properties.

Another research involving various GO formulations functionalized with methacryloyl, zinc ions, and phenylboronic acid [PBA) fabricated onto titanium substrates exhibited superior antibiofilm and antimicrobial activities compared to native titanium [178]. Cells were observed to clump together with visible damage or total lysis. As noted in previous studies, the ROS generated was higher than the control and high levels of DNA were detected spectrophotometrically. This suggested that the antibiofilm and antimicrobial activities associated with this conjugate is likely due to the generation of ROS and damage to the cell membrane.

Reduced GO functionalized with curcumin (rGO-Cur-PDI) significantly reduced biofilm at a MBIC of 125 µg/mL with an LED irradiation time of 60 s [179]. Furthermore, rGO-Cur-PDI was able to downregulate the expression of the genes *efa*, *esp*, *gel*, and *fsr*, which are involved with biofilm formation and growth. The combination also increased the intracellular ROS level by 8.3 folds. In a similar study, silver synthesized on an aqueous GO (Ag-GO) demonstrated superior biofilm reduction compared to the control, 2.5% sodium hypochlorite [180].

On another note, GO was utilized in vitro to determine their ability in osteogenesis of contaminated titanium implants [181]. The authors concluded that brushing and GO at 256 µg/mL was the ideal concentration as higher dosages have previously been reported to exert toxicity towards certain cell lines.

Antibiofilm and antimicrobial activities observed with graphene have been attributed to direct interaction or contact and the generation of oxidative stress. Graphene reportedly exerts the former by wrapping themselves around targeted cells [182] or through their nanostructure sharp edges, which allows them to penetrate and disrupt the cell membranes [183]. The mode of action is dependent on the type of graphene as smaller GO have been observed to form nanopores through their sharp edges which causes intracellular leakage and eventually cell death whereas, bulk GO sheets were observed to induce ROS overproduction and charge transfer mechanisms thus, causing oxidative stress [184]. The functional groups attached to graphene play a role in determining its characteristics such as in the case of GO, which enhanced bacterial adhesion [174] when compared to GNP. Previous findings suggest that graphene could provide a new direction for future prophylactic therapy [176] against persistent infections.

#### 5.2.6. Nanodiamonds

Nanodiamonds (NDs) are carbon nanoparticles with a truncated octahedral architecture that measures between 2 to 8 nm in diameter [185,186]. They consist of a highly ordered diamond core surrounded by a layer of functionalized groups that serve to stabilize the particles. They are promising drug delivery vehicle for a wide range of therapeutics due to their physiochemical properties such as small particle size, purity, high biocompatibility, photoluminescence, and inexpensive synthesis. In comparison to QDs, NDs are considered the non-toxic alternatives for imaging purposes [187].

NDs exhibit surface properties and non-toxic nature that can be manipulated to improve the effectiveness of their delivery characteristics, for instance, their intracellular release of substances, which sets them apart from other carbon-based materials. They are highly stable in corrosive media such as low stomach pH compared to metal or metal oxide nanoparticles which would reduce the risk of their decomposition into toxic materials with decreased activity [188]. Many of the research studies on NDs investigated the possibility of integrating them into improving dental resins.

Properly functionalized NDs have been observed to exert antimicrobial properties similar to those reported with CNTs, whereby they increase the hydrophilicity of surfaces to prevent microorganisms from adhering to those surfaces, hence preventing biofilm formation. Trithiomannoside cluster conjugated to NDs (ND-Man_3_) exhibited potent inhibition of *E. coli* adhesion to yeast and biofilm cells, which subsequently inhibits biofilm formation [189]. The conjugate was able to inhibit the adhesion to yeast cells at a titer of 3.14 µg/mL, which was reported to be 91 times more efficient than the unconjugated NDs. Their antibiofilm activity was amplified 133 times compared to their unconjugated counterparts.

As NDs tend to aggregate to form clumps, quarternized poly (4-vinylpyridinium-co-2-hydroxyethylmethacrylate) were used to chemically functionalize NDs to develop a novel resin-based dental material that exhibit antimicrobial properties with improved mechanical strength. The combination seems to exert an antibacterial effect in a dose-dependent manner as the number of *S. mutans* ATCC 25175 cells that adhered, and area of viable cells reduced with increasing concentration. The mechanism behind this has been associated with its hydrophilic characteristics which leads to the formation of a hydration layer that prevents the adherence of hydrophobic microorganisms whilst inhibiting the formation of biofilms by adhered bacterial cells [190].

Besides that, NDs incorporated onto poly (methyl methacrylate) (PMMA) nanocomposite have been reported to exhibit improved mechanical properties and fungal resistance [191]. The combination reduced the CFU count of viable *C. albicans* KCOM 1301 that could adhere to the modified resin as well as decrease the biofilm thickness through the generation of positive zeta potential and increasing hydrophilicity, which would reduce the interaction of the microorganisms with the surface.

Based on the few findings available, NDs can exert antimicrobial properties against both planktonic and biofilm cells through the generation of positive ions on targeted surfaces, which would form a layer that prevents the interaction of microorganisms with surfaces. Additionally, the inclusion of NDs have been observed to enhance the mechanical strength of compounds such as dental resins, enabling them to withstand more strenuous impact. NDs are still relatively new and would require more in vivo data to determine their safety for clinical applications.

#### 5.2.7. Dendrimers

Dendrimers are nanometer sized, multivalent molecules with branched structures. They are made up of a central core, branches and terminal functional group that encloses the surface of the macromolecules and dictates their drug entrapment efficacy [192]. They are viewed as ideal carriers for biomedical applications as their dendritic architecture can be manipulated to suit one’s purpose. The toxicity associated with dendrimers is attributed to the high cationic charges on their periphery, which can lead to membrane disruption. Suggestions to minimize the toxicity includes selecting neutral or anionic dendrimers that are biocompatible and modifying the peripheral charges by chemical modifications [193].

Dendrimer FD2, a multivalent fucosyl-peptide dendrimer completely inhibited the biofilm formation by *P. aeruginosa* at 50 µM without any apparent toxicity towards the bacterial cells [194]. Ge et al. [195] studied the incorporation of 1% poly(amidoamine) dendrimer (PAMAM) and 5% dimethylamonidodecyl methacrylate (DMADDM) to develop novel anti-caries adhesives. The adhesive exhibited the ability to inhibit essential activities such as the production of lactic acid, metabolic activities, and EPS system of 3 common dental caries pathogens, *S. gordonii* DL1, *S. mutans* UA159, and *S. sanguinis* SK1. The combination exhibited strong antibiofilm and antimicrobial activities with remineralization capabilities that could be potentially beneficial for clinical applications. Interestingly, the adhesive aided in the development of healthier biofilms that consisted of lower proportions of the cariogenic *S. mutans*, which reduces the induction of dental caries.

Two peptide dendrimers, G3KL and TNS18, that were tested against multidrug resistant *P. aeruginosa* PA14 exhibited antibiofilm abilities below the MIC, with a stronger activity noted for the latter [196]. Both G3KL and TNS18 were able to inhibit the biofilm formation of *P. aeruginosa* by 50 and 60%, respectively, at 1/8× MIC in a dose-dependent manner. Likewise, G3KL and TNS18 were able to disperse the preformed biofilms by 32 and 55% at 16× MIC, with the lowest number of bacteria and greater reduction of biomass noted for TNS18 through scanning electron microscopy (SEM) and confocal laser scanning microscopy (CLSM). Further examination of the bacterial structure demonstrated that the peptides targeted the lipid bilayer which leads to a loss of membrane potential and eventually, cell death.

Azithromycin-conjugated clustered nanoparticles were prepared electrostatically between PAMAM and 2,3-dimethyl maleic anhydride (DA) modified poly (ethylene glycol)-block polylysine [197]. The conjugates were observed to release small sized (6.5 nm) and positively charged (23.8 mV) azithromycin conjugated PAMAM nanoparticles that facilitated the transportation of the antimicrobial through the biofilm layer and increased their retention capability in an acidic biofilm environment (pH 6.0). Therapeutic effects of the conjugates were verified through the reduced bacterial burden and decreased inflammation in *P. aeruginosa* chronic lung infection model. The authors described the conjugates as a potential antibiofilm nanoplatform to address biofilm-associated infections.

Dendrimers have exhibited antibiofilm and antimicrobial activities against planktonic and biofilm cells. Their incorporation is observed to enhance the properties such as strength of surfaces or compounds. Based on the previous findings, it is likely that dendrimers exert their antibiofilm effects through surface charges to prevent the interaction of microorganisms with surface. Additionally, the antimicrobial effects have been shown to be non-toxic to bacterial cells, which would prevent the development of resistance.

#### 5.2.8. Mesoporous Silica Particles

Mesoporous silica nanoparticles (MSNPs) are 30 to 300 nm in size and consist of hundreds of empty channels arranged in a honeycomb-like porous structure that are able to absorb large amounts of bioactive molecules. They are considered an ideal carrier for targeted drug delivery with controlled release [198] due to their biocompatible and low toxicity [199]. They have also exhibited attractive, tunable characteristics such as size, shape, high stability, and high surface-to-area ratio.

MSNPs were functionalized with silver-indole-3 acetic acid hydrazide (IAAH-Ag) complexes through hydrazone bond to determine the ability of this combination to kill malignant bacteria [200]. The pH-sensitive complex exhibited a concentration dependent inhibitory effect towards *E. coli* and *S. aureus* with an enhanced inhibition towards the latter. Antibacterial activities exerted by MSNPs conjugate towards the test pathogens appears to be a complementary effect of their ability to reduce the amount of genomic DNA produced, the generation of ROS and their ability to facilitate the movement through the complex biofilm structure even at 30 µg/mL.

Biodegradable disulfide-bridge MSNPs were utilized as a drug carrier to deliver silver nanoparticles and chlorhexidine directly to *S. mutans* ATCC 25175 biofilms [201]. The combination induced an increased and prolonged inhibitory effect towards the test pathogen in comparison to free chlorhexidine due to the slow and sequential release of chlorohexidine as well as silver ions in acidic or reducing environments. The combination also exhibited less toxicity towards oral epithelial cells with no observable toxicity in vivo.

A similar finding was reported by Bai et al. [202] who utilized quaternary ammonium silane-grafted hollow mesoporous silica as a sustained delivery system for metronidazole while taking advantage of its contact killing abilities towards the single-species biofilms of *E. coli* ATCC 25923, *P. gingivalis* ATCC 33277 and *S. aureus* ATCC 25923. MSNPs were shown to prolong the drug release resulting in an enhanced antibacterial effect with minimal cell toxicity even at 100 µg/mL.

Malachite green (MG) encapsulated MSNPs (MG-MSNP) for antimicrobial photodynamic inactivation of *E. coli* and *S. aureus* planktonic as well as biofilm cells showed an increased uptake of MG by the microorganisms at 51.40 ± 3.644% and 68.87 ± 2.02% after 180 min, respectively [203]. Biofilm inhibition was noted at 65.68 ± 2.62% and 79.66 ± 3.82% for *E. coli* and *S. aureus* with a significant decrease in cell viability. This preliminary finding points at the possibility of utilizing MSNPs as a carrier to treat infections or to remove biofilms from medical appliances. Current finding concurs with the previous findings by Seneviratne et al. [204] who noted that the encapsulation of chlorhexidine with MSNPs leads to a steady and potent release of chlorhexidine over 6 h.

Besides that, epigallocatechin-3-gallate-encapsulated nanohydroxyapatite/MSNPs exhibited the ability to reduce the biomass of biofilms, lower the cellular metabolic activities and reduce the number of bacterial cells, which has been attributed to its slow yet persistent release of epigallocatechin-3-gallate, Ca and P ions over 96 h [205].

Evidence obtained from previous studies indicate that MSNPs are efficient drug carriers that could facilitate the delivery of drugs through the biofilm layers whilst inducing their own complementary inhibitory effect through contact killing. MSNPs also prolonged the therapeutic effect of the drugs conjugated and reduced the cell toxicity often associated with the required MIC dose. Previous in vivo findings have shown the potential of this nanocarrier, however, more data is required before it can be utilized for clinical studies.

#### 5.2.9. Chitosan-Based Nanoparticles

Chitosan are biocompatible, natural linear polysaccharides [206], which exhibit antibiofilm and antimicrobial properties that are dependent on various characteristics including degree of deacetylation (DD), degree of polymerization and molecular weight (MW) [207]. 

Chitosan nanoparticles (CHNPs) have exhibited antibiofilm activities towards various microorganisms. CHNPs significantly reduced the biomass of 7 days old biofilms of *E. faecalis* ATCC 29212 and *E. faecalis* OG1RF after 72 h at 20 mg/mL [208]. Observation with CLSM confirmed that the biofilms were disrupted 24 h after treatment. These nanoparticles reportedly retained their antibiofilm properties for up to 90 days. Additionally, CHNPs inhibited the biofilms of *E. coli* MTCC 723, *K. pneumoniae* MTCC 109, *P. aeruginosa* MTCC 121, and *S. aureus* MTCC 734 by 85% to 97%. However, it is surmised that these findings require further investigation due to the high concentrations of CHNPs required, which were between 200 to 500 mg/mL [209].

A comparison of the antibiofilm properties exerted by CHNPs and sodium hypochlorite towards *C. albicans* ATCC 60193, *C. krusei* CBS 73, and *C. tropicalis* CBS 94 indicated that whilst there was notable biofilm inhibition, there was no significant difference between the two [210]. CHNPs tested against *A. baumannii* CCUG 61012, a multiresistant *A. baumannii* clinical isolate, *P. aeruginosa* ATCC 10145, *P. aeruginosa* clinical isolate, vancomycin-resistant *E. faecalis* (VREF) BAA-2365 and vancomycin-resistant *S. aureus* (VRSA) ATCC 700699) were reportedly more effective towards resistant strains than reference strains [211].

Ikono and colleagues [212] observed a greater antibiofilm activity towards *C. albicans* ATCC 10231 and *S. mutans* ATCC 25175 after an 18 h incubation with CHNPs compared to 3 h. Likewise, CHNPs inhibited the biofilms of *Pseudomonas* sp. isolated from milk samples of cows diagnosed with bovine mastitis by 88.96 ± 0.35% at 280 µg/mL within 24 h. Interestingly, antibiofilm activities exerted by CHNPs intensified in the presence of thicker biofilms [213].

CHNPs have showed promising results when they are utilized alone, but recent data seem to indicate that a combination with othernanoparticles or compounds might enhance their overall effect. A study by Elshinawy and co-authors [214] utilizing CHNPs alongside silver nanoparticles and ozonated olive oil (O_3_-oil) in vitro on *C. albicans* MTCC 227, *E. faecalis* OG1RF and *S. mutans* ATCC 2419 demonstrated that the combination could inhibit single and mixed species biofilms by 97% and 94%, respectively, compared to silver nanoparticles and O_3_-oil separately. The combination significantly reduced the number of viable cells in 7 days old biofilms by 6-log within 48 h, whereas CHNPs alone required 7 days to exert a 5-log reduction.

CHNPs have also been used in combination with photosensitizer to enhance the effects of aPDT/PDI. Chen and colleagues [215] investigated the effects of CHNPs loaded erythrosine (ER) against *C. albicans* MYA-2836, *P. aeruginosa* ATCC 27853, and *S. mutans* ATCC 25175 biofilms under an irradiation dose of 50 J cm^−2^. *S. mutans* biofilms were completely eradicated after 12 h whilst *C. albicans* and *P. aeruginosa* biofilms were observed to significantly reduce the biofilm by ca. 3.5-log and ca. 2-log, respectively, after 24 h. CHNPs loaded with ER exerted a better antibiofilm activity overall compared to free ER and chitosan alone.

CHNPs functionalized with rose-bengal (CHRBnp) against 21 days old biofilms of *E. faecalis* ATCC 29212 was evaluated by Shrestha et al. [216]. A concentration of 0.1 and 0.3 mg/mL of CHRBnp were tested with irradiation doses of 20, 40, 60, 50 J cm^−2^ and fractionated dosage of 10 and 20 J cm^−2^ twice. Double irradiation at 10 J cm^−2^ at 0.3 mg/mL led to a complete disruption of biofilms and efficiently eliminated biofilms compared to other treated groups.

In another study, CHNPs combined with methylene blue (MB) to photodynamically inactivate 24 h biofilms of MRSA UTMC 1442, MDR *P. aeruginosa, P. aeruginosa* ATCC 27853, and *S. aureus* ATCC 25923, at an irradiation dose of 22.93 J cm^−2^ effectively reduced their growth by 3.17, 2.37, 2.73 and 3.54 log_10_ CFU, respectively [217]. Furthermore, carboxymethyl chitosan nanoparticles (CMCNPs) that were combined with ammonium methylbenzene blue (AMBB) to produce chitosan-ammonium methylbenzene blue nanoparticles (CMC-MBB NPs) could remove 7 days old biofilms of MRSA ATCC 43300, *P. aeruginosa* AS12378, and *S. aureus* ATCC 6538 after irradiation [218]. CMC-MBB NPs also exhibited a slow and sustained release of MBB at pH 7.4, albeit with some cytotoxicity due to ROS generation that would require further investigation.

On another note, indocyanine green (ICG) encapsulated CHNPs exhibited a reduction in clinically isolated *A. baumannii* strains by ca. 55.3% compared to the ICG irradiated group with ca. 46.2%, which was also evident in the disrupted biofilm architecture viewed through SEM [219]. A closer observation of the effects exerted by chloroaluminium phthalocyanine (ClAlPc) encapsulated in CHNPs towards the biofilms of *S. mutans* ATCC 25175 at an irradiation dose of 100 J cm^−2^ through SEM revealed that the cells were irregularly shaped and arranged in shorter chains [220].

Sonodynamic antimicrobial chemotherapy (SACT), also known as sonodynamic therapy (SDT), is an interesting therapeutic that activates sonosensitizers via ultrasound to produce ROS that leads to cell damage [221]. Pourhajibagher and colleagues [222] utilized photo-sonodynamic antimicrobial chemotherapy (P-SACT), which is a type of aPDT in combination with SACT to inhibit the proliferation of periopathogens, *A. actinomycetemcomitans* ATCC 33384, *P. intermedia* ATCC 15033, and *P. gingivalis* ATCC 33277 on titanium dental implants. P-SACT/CHNPs-ICG treated group showed a significantly higher capacity to eradicate biofilms compared to the other treated groups. SEM analysis of the biofilm revealed that P-SACT treatment resulted in mainly deformed and dead cell, which suggests that a synergistic effect is obtained with the treatment of aPDT and SACT.

CHNPs exhibit promising antibiofilm properties towards a wide range of microorganisms in vitro. Their exact mechanism of action is unclear however, it appears to be associated with their ability to disrupt or inhibit biofilm formation. CHNPs also demonstrate enhanced synergistic effects when they are utilized in combination with other nanoparticles or compounds. Previous data indicate that CHNPs are promising nanocarriers that could be utilized to combat biofilms. More in vivo studies are required to determine if the effects reported in vitro are similar.

CHNPs versatility alone or in combination with other compounds make them suitable alternative therapeutics to combat biofilms or as a nanocarrier to deliver treatments directly to the targeted sites (Table 3).

### 5.3. Microtechnology and Biofilms

Microtechnology is defined as a branch of technology that involves the manipulation, observation and production of structures at micrometer dimensions [245]. Microfluidics is an important branch of microtechnology [246] that has been gaining interest for the study and understanding of biofilms. Microfluidics involves the manipulation and control of a small amount of fluids in micrometer sized channels, which can be applied to understand the growth of biofilms on catheters via simulation of the local microenvironment [247]. Previous research utilizing microfluidics on biofilms include the study on biofilm formation and adhesion [248], effects of treatment [249,250], effects of synthetic QS [251], and effects of shear stress [252].

Microsensors such as microelectrodes is another branch of microtechnology that has been gaining interest in the study of biofilms. Microelectrodes are micrometer electrodes [253] that have been utilized to monitor changes in the chemical and physiological parameters in a biofilm microenvironment such as oxygen profiles [254,255], temperature [256], and pH [256,257], which can differ during different developmental stages and upon exposure to chemicals. For instance, Lin and co-authors [258] previously noted respiration changes in *E. coli* biofilms after antibiotics exposure using microelectrodes.

Unlike nanomedicine, there has been no era of “micromedicine” as the advances in microtechnology are often overshadowed by the larger number of advancements in nanotechnology. Recently, there has been more interest in this area with the development of novel techniques such as microneedles, which exhibit promising characteristics for enhanced drug delivery and improved patient compliance [259].

#### 5.3.1. Microparticles for Delivery of Antimicrobials

Microparticles are the most common form of microtechnology utilized for drug delivery. They are defined as particles with diameters of 1 to 1000 μm, excluding their walls and core structure composition. They are suitable candidates for drug delivery as they can stabilize active agents and exhibit varying drug release profiles including controlled release of drugs [260]. Subcategories of microparticles include microspheres (microparticles that are spherical in shape) and microcapsules (microparticles with different wall and core materials).

Microparticles matrix materials are composed of either inorganic or organic materials. Inorganic materials for instance, gold, are utilized for their unique properties such as improved stability and magnetism [261]. Meanwhile, naturally occurring organic polymers such as chitosan and alginate as well as synthetic organic polymers such as carboxylated poly(l-lactide) (PLLA), poly(lactide-co-glycolide) (PLGA) and poly-epsilon-caprolactone (PCL) are often utilized for their good biodegradability and biocompatibility properties [262,263,264]. More recently, hybrid microparticles composed of a mixture of inorganic and organic components have been engineered for improved and increased functionality.

Efficacy of microparticles as potential drug carriers for antimicrobial compounds such as commercially available antimicrobials as well as inorganic and organic compounds including silver ions, chitosan, usnic acid, and totarol were previously evaluated. Studies on microparticles with antibiofilm properties performed in the last decade have been summarized in Table 4.

##### 5.3.2. Novel Microtechnology Approaches with Antibiofilm Properties

The possibility of incorporating antimicrobial agents with microparticles such as microspheres and microcapsules to combat biofilms have garnered interest in this field. Several microtechnology approaches with antibiofilm properties include microdressing, microspray, microrods, microswimmers, and microneedles. More recent approaches such as micro-textured and micro-patterned surfaces have also been studied for their ability to prevent biofilm formation through their surface topography [286]. Although there are limited studies on some of these approaches, the promising findings from the available information point towards the possibility of utilizing these approaches to combat biofilms and AMR. Figure 3 illustrates the main types of novel microtechnologies used to combat biofilms and drug delivery.

##### 5.3.3. Coatings with Microparticles

Titanium is an implant material commonly utilized to mimic implant surfaces. Deposition of coating materials can be performed using various techniques with promising results as observed with matrix assisted pulsed laser evaporation (MAPLE), which allows the deposition of polymeric substances in thin, even films that are heat and chemical sensitive with very limited degradation [287]. Studies have shown that coating titanium materials with antimicrobial compounds contribute to its antibiofilm activities, which is promising for the control of implant-related infections. Table 5 summarizes the coating formulations presently utilized for antimicrobial control.

Grumezescu and co-authors [288] produced UA-loaded microspheres from polylactic acid (PLA) and polyvinyl alcohol (PVA) to form a thin coating around titanium discs. UA has been reported for a variety of biological activities including antimicrobial, anti-inflammatory and anti-proliferative properties [293]. The microsphere coating allowed controlled and extended release of UA, which significantly reduced *S. aureus* biofilms as the coated surface was not suitable for biofilm adhesion and formation. Good biocompatibility was also observed as mesenchymal stem cells (MSCs) developed and adhered normally without inhibitory or cytotoxic effects.

In a follow-up study, Grumezescu and co-workers [289] incorporated iron oxide nanoparticles to produce UA-loaded magnetic PLGA-PVA microspheres that were coated thinly onto titanium discs by MAPLE. Iron oxide nanoparticles was selected as it had demonstrated antibiofilm activity and excellent biocompatibility [294]. The combination inhibited *S. aureus* biofilm formation by affecting adhesion and maturation stages. Additionally, it exhibited good biocompatibility with MSCs. Magnetic property of this combination has the potential to improve antimicrobial drug delivery and prevent biofilm formation.

Grumezescu et al. [290] also studied other antimicrobial compounds in combination with different matrix materials to produce magnetite and eugenol-loaded poly (3-hydroxybutyricacid-co-3-hydroxyvaleric acid) (P(3HB-3HV)-PVA microspheres. Eugenol is an essential oil derived from cloves with antimicrobial properties [295]. The combination inhibited the adherence and biofilm formation of *P. aeruginosa* and *S. aureus* with more effect towards the latter. Cell viability assay involving endothelial cells showed maintained viability compared to the uncoated control slides.

On another note, Jennings et al. [291] produced chitosan-loaded silver-decorated calcium phosphate microspheres that were chemically bonded to titanium to form a coating via alkyloxysilane reaction for the treatment of dental and orthopedic implant infections. Silver and chitosan were utilized as they have been described for their antimicrobial properties with the latter exhibiting controlled release of silver. Meanwhile, the incorporation of chitosan and calcium phosphate promoted osseointegration [296,297]. The combination exerted significant antibiofilm activities towards *P. denticola, P. gingivalis*, and *S. aureus* in a dose-dependent manner with optimum activity observed using 15% silver. Antibiofilm activities were only significant in the presence of silver.

In their latest study, Grumezescu and colleagues [292] investigated their previously produced P(3HB-3HV)–PVA microspheres with different antimicrobial compounds and modifications to the matrix materials. Lysozyme (Lys), an enzyme that plays an important antimicrobial role in innate immunity with potent antibacterial activity was used [298]. Polyethylene glycol (PEG) was added on to the matrix material. Two formulations, P(3HB-3HV)/Lys and P(3HB-3HV)/PEG/Lys microspheres, were coated onto separate titanium surfaces by MAPLE. Both combinations demonstrated significant antibiofilm activities against *P. aeruginosa* and *S. aureus* through the inhibition of bacterial adherence and biofilm formation. The presence of PEG enhanced the antimicrobial activity of P(3HB-3HV)/PEG/Lys as its porous structure allowed the rapid release of lysozyme. Both combinations showed good biocompatibility when tested against human osteoblast-like SaOs2 cells and umbilical vein endothelial cells.

##### 5.3.4. Micro Textured/Patterned Surfaces

Micro-textured and micro-patterned surfaces utilize biomimetics approaches to produce antibiofilm surfaces. Some of the applications for these surfaces include endotracheal tubes (ETT), urinary catheters, and central venous catheters (CVSs), which are utilized to inhibit bacterial attachment, colonization, and biofilm formation without antimicrobials. Previous findings show promising results of incorporating micro-textured or micro-patterned surfaces for medical devices-related infections.

A study performed on silicone urinary catheters (Foley) with micropatterns showed inhibition of *E. coli* colonization and migration assay, which is an important virulence factor for CAUTIs [299]. May and co-workers [300] observed a significant reduction in colonization by 99.9% in *A. baumannii*, *E. coli*, *K. pneumonia*, MRSA, and *P. aeruginosa* on simulated ETT composed of micropatterned silicone surfaces. Further examination of the surface towards biofilm formation of *E. coli* and *P. aeruginosa* saw a significant reduction in the latter.

Micropatterned surfaces exhibited significant reduction in bacterial colonization and transference of *P. aeruginosa* and *S. aureus* in vitro [301]. A closer examination in vivo of *S. aureus* demonstrated a significantly lower bacterial burden in rat models implanted with micropatterned surface compared to smooth surface control. Similarly, Vasudevan and colleagues [302] studied the fouling of urinary catheters by *E. cloacae* using various microscale patterned polydimethylsiloxane (PDMS) surfaces prepared by photolithography. All patterned surfaces significantly decreased bacterial area coverage compared to smooth PDMS, with a cross-patterned surface, C-1 PDMS demonstrating the most reduction in fouling.

Chien and colleagues [303] produced a patterned surface based on the microscale structure of shark skin to understand its antibiofilm properties. Shark skin obtained from the tail, abdomen and pectoral fin were utilized to produce PMMA replicates. Microstructure and topography of replicates were similar to actual shark skin denticles. Replicates notably reduced the biofilm formation of *E. coli* and *S. aureus* after 14 days with the former exhibiting better initial attachment, which has been associated with the large surface area. Findings from this study highlights the importance of surface topographies for initial bacterial colonization and its possible role in medical device-related infections.

A novel technique to develop dental titanium implant surface with antibiofilm properties using laser micro-texturing was performed by Ionescu et al. [304]. Titanium surfaced discs were used to produce laser-treated discs with regularly spaced hemispheric pits of 20 μm diameter with the controls being grit-blasted and untreated titanium disks. Human enamel discs were used as reference and saliva was used to produce an oral microcosm inoculum. The authors observed that the lasertreated discs had the least biofilm formation in vitro while, both laser-treated and untreated dental implants had significantly lower biofilms formation when compared to grit-blasted surfaces. The findings suggest that micro-texturing can affect biofilm formation on dental implants albeit further work is required as there was no significant difference between the laser-treated and untreated implants. Overall, the findings obtained indicate that the area of micro-patterned and micro-textured surfaces is endless and can be further developed to improve their antibiofilm activities.

##### 5.3.5. Microdressing

Microdressing is a promising approach which can be used to promote wound healing and prevent infections. Several studies have combined components with known wound healing properties and microtechnology to enhance wound healing and antimicrobial properties. Bayón and co-workers [305] produced a silver phosphate bacterial cellulose (AgP-BC) scaffold by synthesizing silver phosphate microparticles (AgPMPs) onto bacterial cellulose membrane surface to allow the latter to exert its wound healing properties whilst preventing infections. Bacterial cellulose is a biopolymer that can be used for wound dressings [306] as it functions to enhance granulation, gas exchange, maintain wound moisture, and absorb wound exudate [307]. However, the presence of warmth and moisture also creates a suitable environment for microbial growth. Incorporation of AgP-BC scaffold enhanced the loading capacity of ciprofloxacin by 6-folds. The synergistic effect between silver and ciprofloxacin resulted in the effective killing of *E. coli* and *S. aureus* biofilms and an extended drug release profile.

Similarly, Thinakaran and co-researchers [308] produced a potential wound dressing substrate, micro fibrous polycaprolactone (PCL) mats via facile centrifugal spinning system with the electrophoretic deposition (EPD) of chitosan and PEG along with silver nano particles (AgNPs) to form polymeric microspheres on PCL. PCL is a widely used synthetic biopolymer [309] and fibrous membrane that is versatile to wound shapes and has a large volume for absorption as well as gas exchange [310]. Chitosan was added to improve hydrophilicity and moisture retention while PEG was used as a stabilizer in the synthesis of AgNPs. The combination inhibited the growth of *P. aeruginosa* and *S. aureus* biofilms but was unable to effectively disrupt mature biofilms. Electrically activated AgNPs was able to effectively inhibit matured biofilms. Electrical activation has also been associated with enhanced wound healing and increased skin perfusion [311]. The combination showed good biocompatibility as it exerted acceptable haemolysis ratios with limited cytotoxicity.

##### 5.3.6. Microspray

Microspray involves the removal of biofilms via shear stress by high-velocity molecules. Most studies on this area have reported the potential of this method in the removal of oral biofilms. Rmaile et al. [312,313] demonstrated that the exposure of *S. mutans* biofilms to high-velocity water microsprays using a prototype Philips Sonicare AirFloss resulted in biofilm detachment in interproximal (IP) spaces. Moreover, Fabbri et al. [314] observed a reduction in biofilm thickness, biomass, and area coverage of *S. mutans* when exposed to high-velocity water microspray using a commercialised oral care device, Philips Sonicare AirFloss in a different in vitro IP model. In their follow-up study, Fabbri and co-authors [315] incorporated fluorescent microbeads to the high-velocity water microspray containing antimicrobials, 0.2% chlorhexidine (CHX), or 0.085% cetylpyridinium chloride (CPC) solutions to observe the penetrative effects towards *S. mutans* at a 30° or 90° angle. Microspray method effectively delivered the microbeads and antimicrobials into the biofilms at a 30° angle.

##### 5.3.7. Microrods

Mair et al. [316] utilized magnetically rotated microrods to disrupt *A. fumigatus* biofilms and enhance the antifungal activity of AMB. Gold-iron-gold (Au-Fe-Au) microrods were approximately 1.2 μm long and the presence of iron allowed magnetic rotation. Efficacy of the microrods and AMB were tested independently and in combination against *A. fumigatus* biofilms. No significant reduction in CFU was observed across treatment groups compared to the controls except for the combination which exhibited more than a 90% kill rate. Microrods affected biofilm integrity by colliding with the fungal hyphae whilst the AMB released killed fungal cells during magnetic rotation. Due to the limited studies on microrods, future studies that include biocompatibility are required.

##### 5.3.8. Microswimmers

Houry and co-researchers [317] utilized flagella propelled bacterial swimmers to effectively disrupt *S. aureus* biofilms by tunneling through the matrix, which allows better penetration of antimicrobial. On another note, Stanton and co-workers [318] combined non-pathogenic magnetotactic bacteria, *Magnetosopirrillum gryphiswalense* (MSR-1) to ciprofloxacin (CIP)-loaded mesoporous silica microtubes (MSMs) to produce MRS-1-CIP-MSM biohybrid system to target *E. coli* biofilms. MSR-1 was utilized for its bipolar flagella, which allows movement and its ability to react to magnetic fields. CIP is only soluble at low pH, which is similar to the microenvironment in biofilms. Biohybrid was guided into the biofilm where it eventually remained trapped, which would be advantageous for drug delivery. MSR-1 exerted antimicrobial properties that have been associated with physical damages to the biofilm matrix. Biohybrid demonstrated optimum antibiofilm activity at 48 h.

##### 5.3.9. Microneedles

One of the major issues of current antimicrobials in treating biofilms are their limited penetrative ability. Xu et al. [319] produced self-dissolvable polyvinylpyrrolidone (PVP) microneedle patches with chloramphenicol (CHL)-bearing and gelatinase-sensitive gelatin nanoparticles (CHL@GNPs) loaded needle tips that could penetrate the biofilm matrix of *Vibrio vulnificus* through physical disruption and deliver antimicrobials directly into the biofilms. High levels of gelatinase are found in chronic wounds [320], which will aid the release of CHL. Microneedles penetrated the biofilm matrix and dissolved to release the nanoparticles, which reacted to the gelatinase produced by microbial communities and led to the release of CHL. A closer examination using a fluorescent labelled dye indicated that the combination could cause biofilm disruption and effective diffusion into the biofilm. The combination patch also significantly decreased the CFU/mL at 4 to 8 h of incubation with decreased cytotoxicity towards fibroblast cells and improved cell proliferation.

Similarly, Permana and co-authors [321] produced bacterial sensitive nanoparticles loaded with doxycycline (DOX-NP) and decorated with chitosan, which were incorporated into dissolving microneedles for the treatment of bacterial biofilm skin infections primarily, in burn and chronic wounds. PLGA and PCL were used as the nanoparticles polymer matrices with various formulations produced while the microneedles matrixes were PVP and PVA-based. Chitosan is positively charged and would serve as a targeted drug delivery system as it would be attracted by the negative charges on the surface of the biofilm EPS and bacterial cell walls. The combination showed enhanced antibiofilm and antimicrobial activity towards *P. aeruginosa* and *S. aureus.* Ex vivo studies indicate that microneedles improved penetration ability of the NPs with more than a 97% decrease in bacterial bioburden. This finding was enforced by the negligible release of DOX from nanoparticles in non-infected skin model.

## 6. Conclusions

Most studies for micro- and nanotechnology are in the in vitro stages but they show promising potential for the control and possibly, the eradication of bacterial and fungal biofilms. They exhibit improved efficacy and reduced toxicity with a slim chance of inciting a resistance in targeted microorganisms, particularly when used in combination with antimicrobials. Further studies would be required to ensure the safety aspect of these delivery systems before they can be utilized in a clinical setting. Nonetheless, the data from relevant studies that were covered in this review strongly indicate that utilization of these micro- and nanotechnology approaches will greatly improve the treatment and management of biofilm-related infections. These advances in technology constitute a formidable and promising antibiofilm arsenal to combat AMR in the near future.

## Figures and Tables

**Figure 1 molecules-26-01870-f001:**
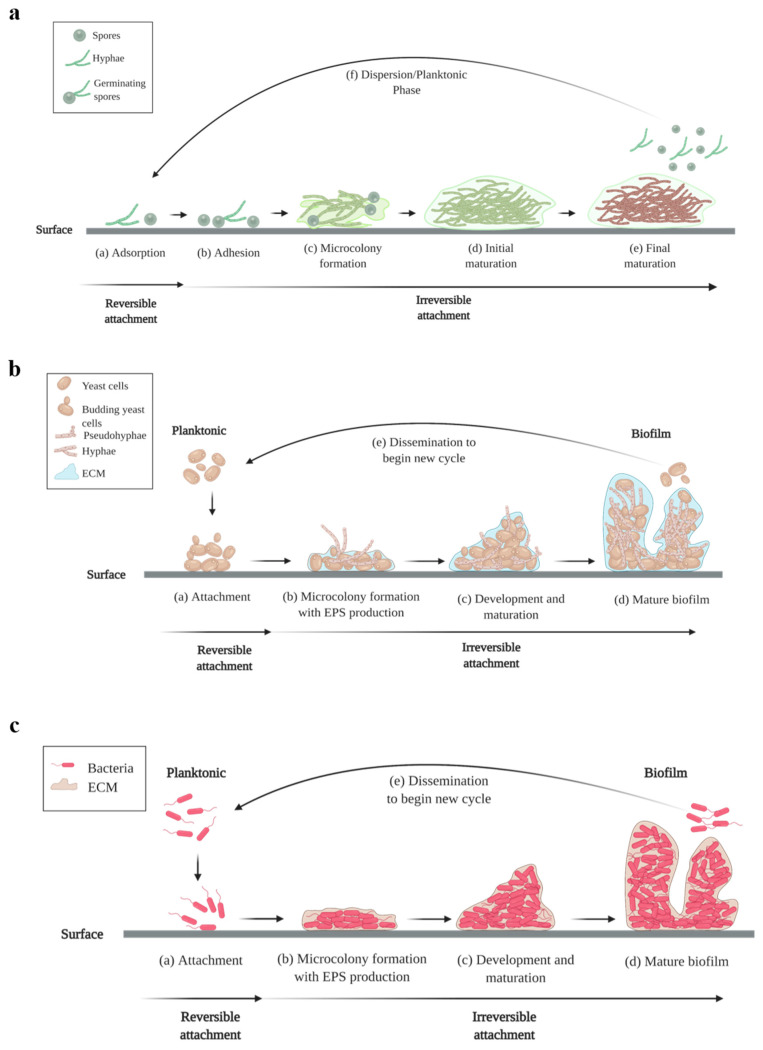
Schematic models of various stages of biofilm development in (**a**) filamentous fungi, (**b**) yeasts, and (**c**) bacteria. Biofilm formation by these three microorganisms share similar processes in terms of their initial attachment to surfaces, the formation of microcolonies, their maturation and dispersal into the surrounding. Initial adhesion is the only reversible step for all these microorganisms.

**Figure 2 molecules-26-01870-f002:**
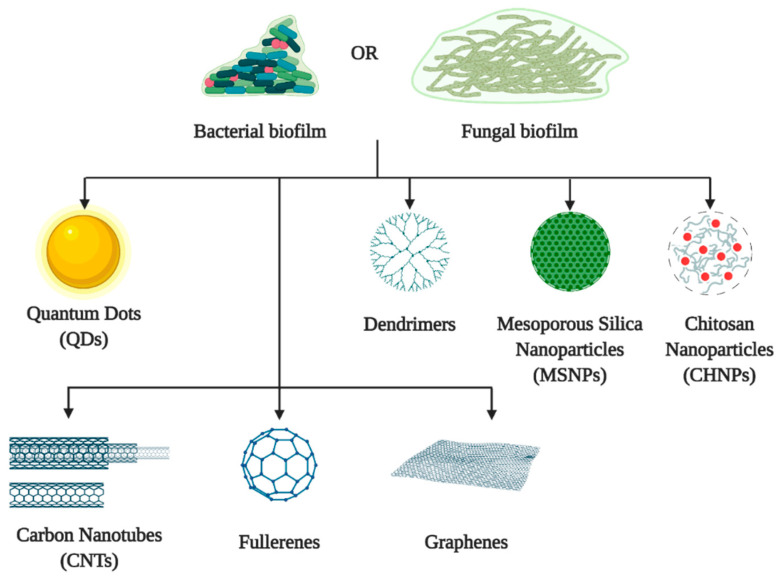
Novel nanomaterials to combat biofilms and as potential drug delivery systems.

**Figure 3 molecules-26-01870-f003:**
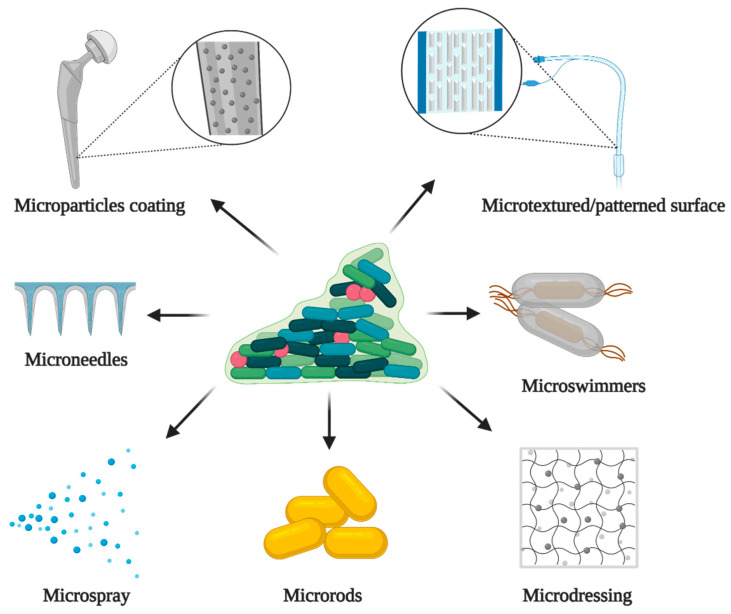
Novel microtechnology approaches to combat biofilms and as drug delivery systems.

**Table 1 molecules-26-01870-t001:** Fungal biofilms and their matrix components.

Species	Matrix Components	References
***Candida* species:**		
*Candida albicans*	Glucose, DNA, small amounts of hexosamine, small amounts of protein, phosphorous, and uronic acid.	[35]
b. *Candida auris*	Polysaccharide complex of mannan and glucan.	[36]
c. *Candida glabrata*	High concentration of carbohydrate and less protein than *C. parapsilosis* biofilms.	[37]
d. *Candida parapsilosis*	High concentration of carbohydrates with less protein.	
e. *Candida tropicalis*	Hexosamine, small amounts of protein, phosphorous, and more uronic acid than *C. albicans.*	[35]
***Cryptococcus* species:**		
*Cryptococcus neoformans*	Glucurunoxylomannan and sugars such as xylose, mannose, glucose, and galactoxylomannan.	[38]
***Aspergillus*** **species:**		
*Aspergillus fumigatus*	Galactomannan, β-1,3 glucan, monosaccharides, galactose, polyols, melanin, a small amount of protein, *N*-acetyl-galactosamine (galactosaminogalactan [GAG]) and eDNA.	[39,40,41]

**Table 2 molecules-26-01870-t002:** Efflux pump encoding genes in fungal biofilms.

Species	ABC Transporters Genes	References	MFS Transporters Genes	References
***Candida* species**				
*Candida albicans*	*CaCDR1* *CaCDR2*	[56]	*CaMDR1*	[56]
b. *Candida glabrata*	*CgCDR1* *CgCDR2*	[57]	-	-
	*SNQ 1*	[58]	-	-
c. *Candida parapsilosis*	*CDR*	[59]	*MDR*	[59]
d. *Candida tropicalis*	*CtCDR1*	[30]	*CtMDR1*	[30]
***Aspergillus* species**				
a. *Aspergillus fumigatus*	-	-	*MDR*	[30]

**Table 3 molecules-26-01870-t003:** Antibiofilm activities of compounds entrapped/immobilized in/on chitosan nanoparticles (CHNPs).

Chitosan Nanoparticles (CHNPs) Based Formulation	Microorganism(s) Tested	References
Vancomycin-loaded Carboxymethyl chitosan-2,2′-ethylenedioxy bis ethylamine-folate nanoparticles	*S. aureus*	[223]
β-*N*-acetyl-glucosaminidase immobilized linoleic acid carboxymethyl chitosan nanoparticles	*S. aureus*, *S. epidermidis*, *A. actinomycetemcomitans*	[224]
Ferulic acid-encapsulated CHNPs	*C. albicans*	[225]
RBRBR-CN (Potent ultrashort antimicrobial peptide—RBRBR)	*S. aureus*	[226]
Ciprofloxacin-loaded CHNPs	*S*. Paratyphi A	[227]
Fucoidan coated ciprofloxacin-loaded CHPNs	*S*. Paratyphi A	[227]
Curcumin-loaded CHNPs	*S. mutans*	[228]
Chitosan-propolis nanoparticles	*E. faecalis*	[229]
Oxacillin and Deoxyribonuclease I-loaded CHNPs	*S. aureus*	[230]
Clove oil-loaded CHNPs	*E. coli*	[231]
Ferulic acid encapsulated chitosan-tripolyphosphate nanoparticles	*P. aeruginosa*	[232]
Cinnamaldehyd- encapsulated CHNPs	*P. aeruginosa*	[233]
Co-amoxiclav embedded CHNPs	*S. aureus*	[234]
Alginate lyase immobilized low molecular weight CHNPs	*P. aeruginosa*	[235]
Chitosan-propolis nanoparticles	*S. epidermis*	[236]
Alginate lyase functionalized CHNPs of Ciprofloxacin	*P. aeruginosa*	[237]
Chrysin-loaded CHNPs	*S. aureus*	[238]
*Tityus stigmurus* Hypotensin-loaded cross-linked CHNPs	*C. tropicalis*, *C. krusei*, *C. albicans*	[239]
Chitosan-propolis nanoparticles	*E. faecalis*	[240]
Mesenchymal stem cells derived conditioned media incorporated CHNPs	*V. cholerae*	[241]
Cellobiose dehydrogenase and deoxyribonuclease I co-immobilized CHNPs	*C. albicans, S. aureus* (Mono- and polymicrobial)	[242]
Glucose oxidase immobilized CHNPs	*S. aureus*	[243]
Curcumin-loaded CHNPs	*C. albicans, S. aureus* (Mono- and polymicrobial)	[244]

**Table 4 molecules-26-01870-t004:** Summary of studies on microparticles with antibiofilm properties.

Formulation	Matrix Material(s)	Active Ingredient(s)	Microorganism(s) Tested	References
**Organic Polymeric Microparticles**				
UA-loaded CPLLA MP	Carboxylated poly(l-lactide)	Usnic acid (UA)	*S. epidermidis*	[265]
DAP-loaded PCL MP	Poly-epsilon-caprolactone	Daptomycin	MRSA, *S. epidermis*	[266]
Ciprofloxacin-loaded PLGA MP	PLGA	Ciprofloxacin	*P. aeruginosa,* *S. aureus*	[267]
PTC-loaded Man-Cyst MP	Mannitol	Polyanion tobramycin complex, Cysteamine	*P. aeruginosa*	[268]
ISMN-loaded PLGA MP	PLGA	Isosorbide mononitrate	*S. aureus*	[269]
DAP-loaded PMMA-EUD MP	Poly (methyl methacrylate) -Eudragit RL 100	Daptomycin	MRSA, *S. epidermis*	[270]
DAP-MP	Poly (methyl methacrylate)-Eudragit RL 100	Daptomycin	MRSA	[271]
PBMP-coated PLGA MP	Poly (butyl methacrylate-co-methacryloyloxyethyl phosphate), PLGA	Furanone C-30	*S. mutans*	[272]
Chitosan MP	Chitosan	Chitosan	*S. mutans*	[273]
**Inorganic Microparticles**				
CHX-loaded Ca(OH)_2_ MP	Calcium hydroxide	Chlorhexidine	*S. mutans*	[274]
**Hybrid Microparticles**				
Mino-Ca-DS MP	Calcium chloride, dextran sulfate	Minocycline	*A**. actinomycetemcomitans*, *S. mutans*	[275]
SNO MP	Porous organosilica	Nitrosylated thiol groups	*P. aeruginosa*	[276]
**Organic Polymeric Microspheres**				
Tetracycline-loaded chitosan MS	Chitosan	Tetracycline, chitosan	*P. aeruginosa*	[277]
MCP MS	Chitosan, Pluronic^®^ F127	Melatonin, chitosan	MRSA	[278]
Totarol-loaded PLGA MS	PLGA	Totarol	*S. aureus*	[279]
Chitosan-alginate MS	Chitosan, alginate	Chitosan	*E. faecalis,* *P. aeruginosa, P. vulgaris, S. aureus*	[280]
RIF-MOX PLGA MS	PLGA	Rifampicin, moxifloxacin	*S. aureus*	[281]
**Inorganic Microspheres**				
Gentamicin-loaded MCH MS	Mesoporous carbonated hydroxyapatite	Gentamicin	*S. epidermidis*	[282]
Gentamicin-loaded MEH MS	Magnetic mesoporous carbonated hydroxyapatite	Gentamicin	*S. epidermidis*	[283]
**Hybrid Microspheres**				
Ag–HA–Alb MS	Hydroxyapatite, albumin	Silver	*S. aureus*	[284]
**Organic Polymeric Microcapsule**				
*Lactobacillus rhamnosus* GG alginate-chitosan MC	Alginate, chitosan	*L. rhamnosus* GG	*E. coli*	[285]

Abbreviations: MP, microparticle; MS, Microsphere; MC, Microcapsule; UA, Usnic acid; CPLLA, carboxylated poly(l-lactide); DAP, Daptomycin; PCL, Poly-epsilon-caprolactone; PTC, Polyanion tobramycin complex; ISMN, Isosorbide mononitrate; PMMA-EUD, Poly(methyl methacrylate)-Eudragit RL 100; PBMP, Poly(butyl methacrylate-co-methacryloyloxyethyl phosphate); CHX, Chlorhexidine; Ca(OH)_2_, Calcium hydroxide; SNO, Porous organosilica containing nitrosylated thiol groups; Mino-Ca-DS, Minocycline-calcium-dextran sulfate; MCP, Melatonin-loaded chitosan/Pluronic^®^ F127; RIF-MOX, Rifampicin-moxifloxacin; MCH, Mesoporous carbonated hydroxyapatite; MEH, Magnetic mesoporous carbonated hydroxyapatite; Ag–HA–Alb, Silver–hydroxyapatite–albumin.

**Table 5 molecules-26-01870-t005:** Summary of coatings used with antibiofilm properties.

Coating Formulation	Coating Surface	Coating Method	Microorganism(s) Tested	References
UA loaded-PLA-PVA MS	Titanium	MAPLE	*S. aureus*	[288]
UA loaded-Magnetic PLGA-PVA MS	Titanium	MAPLE	*S. aureus*	[289]
Magnetite and eugenol loaded P(3HB-3HV)-PVA MS	Glass	MAPLE	*P. aeruginosa, S. aureus*	[290]
Chitosan loaded with silver-decorated calcium phosphate MS	Titanium	Alkyloxysilane	*P. denticola, P. gingivalis, S. aureus*	[291]
P(3HB-3HV)-PEG-Lys MS	Titanium	MAPLE	*P. aeruginosa, S. aureus*	[292]
P(3HB-3HV)-Lys MS				

Abbreviations: Ref, Reference; MS, Microsphere; MAPLE, Matrix assisted pulsed laser evaporation; UA, Usnic acid; PLA, Polylactic acid; PVA, Polyvinyl alcohol; PLGA, Poly(lactide-co-glycolide); P(3HB-3HV), Poly(3-hydroxybutyricacid-co-3-hydroxyvaleric acid); PEG, polyethylene glycol; Lys, Lysozyme.

## Data Availability

Data sharing not applicable to this article as no datasets were generated or analysed during the current study.

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
