# Peer review of "Approaches for Mitigating Microbial Biofilm-Related Drug Resistance: A Focus on Micro- and Nanotechnologies"

_molecules, 2021, doi:10.3390/molecules26071870_

Round 1
Reviewer 1 Report
Dear Authors,
I carefully read the manuscript. It is not always easy to follow the reading of this review. It appears a bit confusing, including some topics that do not seem fully relevant. Some comments are given below.
- The title of the manuscript exploits some ambiguity in the term "controlling". It would seem to be used in the sense, appropriate for a review, of "regulating", but it seems that the authors intended it to mean "monitoring" so that they included sections such as the one on QDs and the one on NDs. I suggest adjusting the content of the manuscript to the title and to the abstract and removing the sections not concerned with the effect of regulating biofilm formation.
- Figures 1, 2 and 3 are redundant; the systems are very similar. Moreover, the images shown appear to be taken from papers. If so, they should be cited in the text and in the references.
- Line 112: correct 1 x 103 in 1 x 103
- Paragraph 2 covers conventional antifungal agents, but in the following the work focuses on antimicrobial agents, not only antifungals.
- Section 3.5 is not correctly formatted.
- Section 4, Types of Nanomaterial, is too general. Describing all types of nanomaterials is beyond the scope of the review. It would be better if the authors focused only on the types of nano (and micro) materials actually used in combating biofilms.
- The section 2. Carbon-based Nanoparticles, in its present form, is out of the scope, I would suggest eliminating it.
- The Conclusion section is not very clear to me. The sentence: “the available data strongly indicates that the utilization of these nanoparticles will greatly improve the treatment and control of biofilm-related infections” is very limiting. It only mentions nanoparticles, whereas the review talks about several other systems. I suggest expanding the comments to include all the systems analyzed.
Author Response
Please seen the attachment file.

Reviewer 2 Report
The manuscript reviewed recent advances in micro- and nanotechnology for controlling microbial biofilms. The authors provide plentiful references and organize some important informations and literatures into tables, which allows readers easier to sort through a lot of information quickly. The paper is well organized and written, and the topic is of great interest. My comments are shown below.
- This paper lacks a literature review related to the important antibacterial material chitosan. There many studies focus on the development of chitosan-based nanoparticles (https://doi.org/10.1016/j.carbpol.2020.116312; https://doi.org/10.1016/j.carbpol.2020.116075; https://doi.org/10.1016/j.carbpol.2020.116254; https://doi.org/10.1016/j.ijbiomac.2020.04.090 https://doi.org/10.1016/j.carbpol.2018.03.101; https://doi.org/10.1016/j.carbpol.2018.03.088; https://doi.org/10.1016/j.msec.2019.109885; https://doi.org/10.1016/j.carbpol.2019.02.018; https://doi.org/10.3390/nano8020088; https://doi.org/10.1016/j.carbpol.2015.02.068; https://doi.org/10.1016/j.carbpol.2018.07.072; https://doi.org/10.1016/j.ejps.2018.01.046), as well as chitosan/inorganic compound (Ag, ZnO, CuO, etc.) hybrid nanoparticles (https://doi.org/10.1016/j.carbpol.2019.03.062; https://doi.org/10.1016/j.carbpol.2019.02.005; https://doi.org/10.1016/j.carbpol.2018.07.039; https://doi.org/10.3390/ijerph16040598; https://doi.org/10.1016/j.msec.2018.10.012; https://doi.org/10.1016/j.ijbiomac.2020.03.077) for antimicrobial and antibiofilm applications. Please cite them in the manuscript to provide important information for readers.
- The font size of the words must be consistent (Table 1-3).
- Use superscript in some text, e.g. Ca2+.
- The name of microorganism must be in italics.
5.: The format of the text in Section 3.6 is changed.
6.Incorrect reference format: reference 252-254
Reviewer 3 Report
The authors of the review entitled “Novel Micro-and Natotechnology for controlling Microbial Biofilms” aimed to provide an overview of the general principles of clinically important microbial biofilm formation as well as the recent approaches that have been developed to combat biofilm formation and its application in clinical settings. Globally, the work is very updated and of general interest for the journal readers. However, there are some issues that should be addressed.
Major comments
- The authors aimed to focus on fungal biofilms, specially in the “ introduction” section. However, table 3 and 4 are only related to bacteria. In my opinion, the authors should include in all introduction sections a brief updated information regarding bacteria biofilms.
Minor comments
- Line 98, 100, 102, 111,… Species names should be in italic. Please revise. throughout the manuscript.
- Table 1 – species name should not be separated.
- In section 2, please delete “conventional”.
- Section 3.6. should not be in italic.
Author Response
Thank you for your comments. We have endeavoured to amend the manuscript quite substantially in line with not just your comments but also those of the other 2 reviewers.
Please see the attached file.

Round 2
Reviewer 1 Report
After a careful re-reading, the work is now quite improved compared to the previous version.
I therefore suggest that it is approved for publication